

# Single-cell analysis of *Mycobacterium tuberculosis* with diverse drug resistance using surface-enhanced Raman spectroscopy (SERS)

Andrey Zyubin[1], Anastasia Lavrova[2,3], Marine Dogonadze[2], Evgenii Borisov[4] and Eugene B. Postnikov[5]

[1] Immanuel Kant Baltic Federal University, Kaliningrad, Russia
[2] Saint-Petersburg State Research Institute of Phthisiopulmonology, Saint-Petersburg, Russia
[3] Immanuel Kant Federal University, Kaliningrad, Russia
[4] Saint-Petersburg State University, Saint-Petersburg, Russia
[5] Kursk State University, Kursk, Russia

## ABSTRACT

In this work, we investigated individual bacteria *M. tuberculosis* belonging to strains of the Beijing family with different drug sensitivity (sensitive, multi and extensive drug-resistant) by surface-enhanced Raman spectroscopy (SERS) in the fingerprint region. The latter is focused on the spectral bands, which correspond to a set of glutathione bands and DNA methylation patterns revealed due to 5-methylcytosine spectral biomarkers. It is shown that these spectral features can be correlated with drug sensitivity and DNA methylation. Thus, since this kind of diagnostics is fast and operates with individual cells, it can be considered a promising tool, which significantly shortens the time required for a strain's type identification necessary to prescribe adequate therapy.

## INTRODUCTION

Drug resistance of pathogens belonging to the genus *Mycobacteria* continues to be one of the critical problems of the public health system, see the works by *Mabhula & Singh (2019)*, *Singh et al. (2020)* for the review of the recent state of the art, As noted in the World Health Organization's Global Tuberculosis Report 2023 GTR (*World Health Organization, 2023*), the amount of multidrug-resistant and rifampicin-resistant cases (MDR/RR-TB) consisted about 3.3% among people with primary diagnosed tuberculosis and reached 17% among those who were previously treated. Lower treatment success is typical when the worst case of extensive drug resistance (XDR) develops. Due to the complexity and costs associated with current approaches to diagnosing drug resistance in Mtb, there is a demand to develop new, faster and more affordable methods for testing drug resistance.

The conventional approaches to this problem include culture growth, methods of molecular diagnostics, and sequencing techniques. However, each of them has limitations (*Koch, Cox & Mizrahi, 2018*). In particular, the growth of mycobacterial cultures under the

Corresponding author
Andrey Zyubin, azubin@mail.ru

action of a set of different antibiotics takes a long time, up to a month, and exhibits some uncertainty when working with clinical strains. Molecular diagnostic methods offer a faster approach and can elucidate the underlying mechanisms of drug resistance. However, these methods often demand costly and specialized kits alongside specific sample collection and preparation protocols, such as the need for culture growth in smear-negative specimens, *etc.*, see the reviews by *Nguyen et al. (2019)*, *Chopra & Singh (2020)*.

Among the prospective methods, which gained attention recently, is Raman spectroscopy, especially surface-enhanced Raman spectroscopy (SERS). A principal advantage of this method is its speed, with single runs completed in minutes, offering significant time-saving over molecular-based techniques that require several hours (*Dhankhar, 2018*). Additionally, the required instrumentation is readily accessible, with portable options available for field use. This method also provides high-resolution molecular analysis and enables the examination of individual bacteria as well as cultures, as noted in studies by *Stöckel et al. (2017)*, *MacGregor-Fairlie et al. (2020)*, *Usman et al. (2023)*.

Due to the complexity of Raman spectra in a wide range of frequencies, one needs to reveal fingerprint ranges, which are most sensitive to the target parameters of identification. Often, this goal can be archived using different methods of machine learning. In particular, *Ho et al. (2019)* demonstrated that deep learning could achieve high identification accuracy at the isolate level for non-tuberculosis bacteria. Furthermore, *Wang et al. (2022)* addressed the classification problem by proposing specific bandwidths as markers. These markers correspond to metabolite constituents that are highly sensitive to resistance status in both pulmonary and extrapulmonary *M. tuberculosis* strains, as well as to those distinguishing these specimen types. In *Ogunlade et al. (2023)*, three bands were revealed as exhibiting variability between wild-type bacteria and four resistant mutants (discussed, respectively, to mutations in the katG gene) explored with the singe-cell-based SERS. Special attention to variations in nucleic acid content considered responsible for resistance to the first-line drug rifampicin was analysed with the SERS applied to samples' products amplified with the polymerase chain reaction (PCR) in *Dastgir et al. (2022)*. Thus, the development of this methodology in application to mycobacterial studies allowed to designate the development of the SERS approach as crucial for the "Big 5" antibiotic resistant challenge, which lists *M. tuberculosis* as one of five major threats affecting public health, as focused in the work of *Hassanain et al. (2022)*.

Following the research program mentioned above, in our previous short report (*Lavrova et al., 2023*), we have revealed evidence that the structure of SERS spectra taken from individual mycobacteria is significantly affected by their drug resistance status. In the present study, this feature is explored in more detail from the point of view of molecular biophysics. We analyze SERS spectra from single *M. tuberculosis* cells, focusing on specific spectral band regions obtained at various points across an individual cell. The principal exploration is focused on peak intensities of spectral features associated with lipids and proteins in the spectral region of $400 - 1,800$ cm$^{-1}$. At first, it aims to distinguish between antibiotic sensitivity of strains of *M. tuberculosis* belonging to the Beijing family (it is a typical representative of mycobacteria in Russia) as a feature of their spectral fingerprint

differences. Additionally, the spectral specificity of the considered biomarkers allows the biophysical discussion of the respective specificity for clinical pulmonary and extrapulmonary bacterial strains. The existence of a difference between drug resistivity of these cases has been already mentioned by different research groups in wide-range surveys, see *e.g.*, works by *Lai et al. (2011)*, *Lee et al. (2015)*, *Yablonskii et al. (2016)*. Moreover, our previous research (see the work by *Zyubin et al. (2019*, *2021)* carried out with populations of mycobacteria sampled from different localizations revealed certain differences reflected in Raman spectra.

## MATERIALS AND METHODS

### Microbial preparation

For this study, we used mycobacteria from the library of Mtb clinical strains collected in the Saint-Petersburg State Research Institute of Phthisiopulmonology. Clinical strains were obtained by culturing sputum samples from patients earlier hospitalized at the Institute (see Table 1). All patients had signed a written consent to store anonymized processed clinical material. This material's usage for the study has been approved by the Independent Ethics Committee at the Saint-Petersburg State Research Institute of Phthisiopulmonology. No human participants were involved in this research, which operated with the bacterial strains kept in the Institute's library.

The samples were stored in the library in the frozen state at −80 °C as a dense suspension in a physiological solution with 15% glycerol. The details of procedures of sample collecting and preparing for storage can be found in *Zyubin et al. (2019)* as well as procedures applied to assign them a drug resistance status.

The strains designated for subsequent culturing, deactivation, and examination *via* Raman spectroscopy methods were retrieved from storage and made available as of September 15, 2021. No additional human participants were involved in the present study. Among the bacterial samples extracted from the library (anonymous respectively to the patients' names), we used material of six strains belonging to the following types: (A) strains isolated from the respiratory material of three patients suffering from pulmonary tuberculosis and (B) strains isolated from the site of the bone destruction after the surgical treatment of patients suffering from the bone tuberculosis. The drug sensitivity to particular kinds of medications of the chosen samples was assigned to them in the library's description based on the standard method of the growth quantification on the Löwenstein–Jensen medium and/or by the BACTEC MGIIT 960. Table 1 reports this property of the explored strains; their denotation as multidrug-resistant (MDR) and extensive drug-resistant (XDR) follow the WOS classification.

Following the protocol described in detail earlier (*Zyubin et al., 2019*), the samples were unfrozen suspended in 250 $\mu$l of distilled water, killed by heating at 80 °C during 20 min and re-suspended in 100–150 $\mu$l of distilled water after centrifugation during 20 min at 2,000 rpm. A drop containing diluted bacterial suspension was placed on the SERS substrate's surface using an automated pipette in an amount of 5 $\mu$l. The bacterial content was adjusted in such a way that after drying one can clearly identify one separate bacterial cell in the field of the 100× objective of an optical microscope.

**Table 1  List of the investigated clinical strains of *M. tuberculosis*, Beijing clade.**

| Sample No. | TB type | Localization | Drug resistance | Type of drug sowing | Date of material's freezing | Date of strains' |
|---|---|---|---|---|---|---|
| 8692 | Pulmonary | Respiratory | SENS | | 26.11.2010 | 09.12.2010 |
| 6679 | Extra pulmonary | Surgical | SENS | | 06.09.2011 | 23.09.2011 |
| 1604 | Pulmonary | Respiratory | MDR | S, H, R, K | 30.03.2010 | 09.04.2010 |
| 109 | Extra pulmonary | Surgical | MDR | S, H, R, K, Cp, A | 13.01.2012 | 30.03.2012 |
| 758 | Extra pulmonary | Surgical | XDR | S, H, R, E, Of, Z | 03.02.2010 | 09.04.2010 |
| 9622 | Pulmonary | Respiratory | XDR | S, H, R, E, K, Of, Z | 28.12.2010 | 24.01.2011 |

Note:
SENS, sensitive; MDR, multidrug-resistant; XDR, extensive drug-resistant; S, streptomycin; H, isoniazid; R, rifampicin; E, ethambutol; Z, pyrazinamide; K, kanamycin; Of, ofloxacin; Cp, capreomycin; A, amikacin.

## SERS substrates

Gold SERS substrates were bought from Silmeco (Denmark). They consist of gold-coated freestanding silicon nanopillars and adopted for the $\lambda = 785$ nm excitation. Under this condition, these substrates allow the enhancement of the Raman signal up to $10^4$ times.

## SERS instrumentation

SERS-based records were performed with Raman Senterra (Bruker, Billerica, MA, USA) spectrometer under the following operational conditions: DPSS laser excitation corresponds to 785 nm and generates the power of 50 mW on the sample; the range of the spot size varies ranged from $1 \times 3$ to $1 \times 5$ $\mu$m. Such a spot provides an opportunity for accurate positioning in different points of a single bacterial cell's surface, see Figs. 1B and 2 using a confocal microscope with a 100× (NA 0.9) objective; the diffraction grating corresponds to 1,200 gr/mm$^{-1}$; Rayleigh scattering was eliminated by the notch filters.

The optical scheme was supplied with a CCD camera IDUS 416 (Andor, London, UK) with digital picture resolutions of $1,024 \times 256$ pixels. The preliminary calibration was carried out using a silicon substrate at a static spectrum centred at 520.1 cm$^{-1}$ for 1 s, and the sample control measurements without SERS were carried out for bacterial spectra using chemically purified quartz glass as the substrate. Due to the enhancement of the SERS structure, the surface plasmon field-enhanced Raman spectrum of the bacterial cell was registered from points of a single cell using the Raman Senterra spectrometer.

For each biological replicate per strain, the spectrum was recorded with 70 s acquisition time; the final spectra (one for each position on the cell's surface) are represented by averaging over three registered measurements. Figure 1 illustrates such raw (without extracting auto-fluorescence-based baselines) spectral curves in correspondence to the points of a bacterium, where they were recorded. Such raw spectra were preprocessed before analysis, see below. The total number of measurements contains 172 spectra. Among this set, 60 of them exhibit the best signal-to-noise ratio; they were chosen for the subsequent analysis of the six different strains of Mtb.

## Spectral preprocessing

SERS spectra were recorded within the range of 500–1,800 cm$^{-1}$, which belongs to the so-called "fingerprint" region. All spectra were normalized on the maximum intensity. The
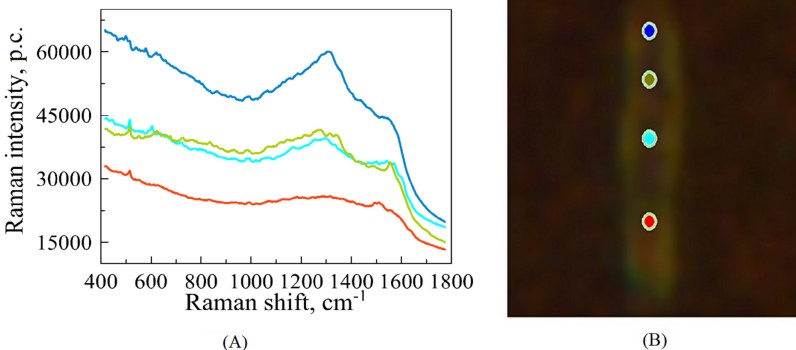

**Figure 1** An illustrative example of raw SERS spectra of Mtb cell (A) taken on Silmeco substrates ("p.c." means photon counts reported by the instrument) and a bacterial image at 100x optical magnification superposed with points indicating the location of spectral measurements (B). The colours of spectral curves in (A) correspond to the colours of points marking the laser positioning (B).

chosen raw spectra were preprocessed by following the standard route with a homemade MATLAB code. The first step consisted of the removal of spike-like artefacts that emerged usually due to cosmic rays. For this goal, the algorithm proposed in the work by *Whitaker & Hayes (2018)* was applied. It reveals the range of sampled wavenumbers, where the indicator, proposed in the cited article exceeds some prescribed limit. Here we used the level, *i.e.*, the tenfold overcoming the median value over the whole sample. In contrast to the approach reported by *Whitaker & Hayes (2018)*, we replaced the raw values within such intervals, not by those obtained with a weighted moving averaging but weighted moving median filter. This replacement is substantiated by the character of noise in the studied biological samples. The length of the median filter was chosen as a minimal odd length, which does not lead to local zero weights. The sample's values, which were not classified as artificial spikes, were kept in the output sample the same as in the input to prevent double data smoothing. The second step consisted of this smoothing carried out with the cubic Savitzky-Golay filter with the window length equal to 21 (when another value is not specified explicitly). This relatively large width was determined as preferred for two reasons: (i) reach high-frequency content of the noise and (ii) the goal to reduce the overall number of peaks for analysis to a relatively few numbers of principal ones, which are repeatable over all spatial locations and have biophysical meaning (see the discussion below). Finally, the baseline correction using the MATLAB function made available by *Al-Rumaithi (2023)*, which is principally based on the algorithm proposed in the article developed in the work by *Schulze et al. (2012)*.

## RESULTS AND THEIR DISCUSSION

### Individual cells

During the experiment, we recorded spectra from sensitive (SENS), multidrug-resistant (MDR), and extensive drug-resistant (XDR) bacterial cells at 100× magnification for pulmonary and extrapulmonary antibacterial strains. The applied experimental configuration allowed recording at several localized points (from three to six locations) of

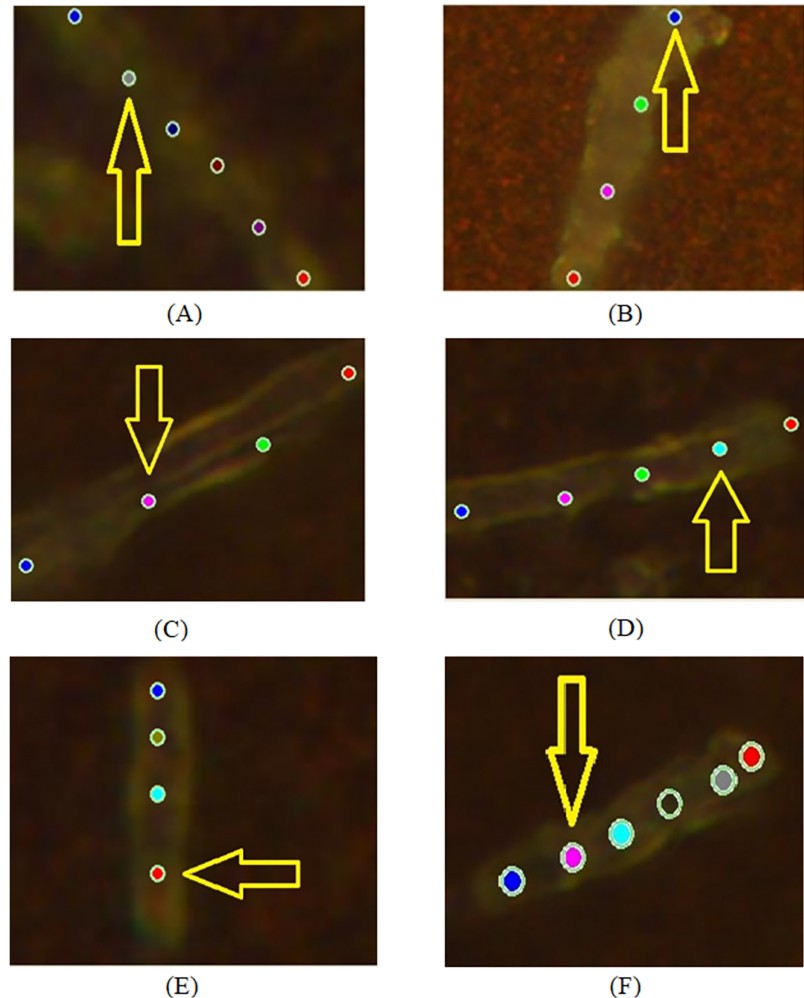

**Figure 2 Examples of images of bacterial cells obtained using 100x optical magnification aimed at showing various locations, where SERS measurements were carried out: (A, B) sensitive (SENS) strains; (C, D) multidrug resistance (MDR) strains; (E, F) extensive drug-resistant (XDR) strains.**

an individual bacterium as shown in Fig. 2. There are the following cases of objects with different drug-resistant status there: SENS (A, B); MDR (C, D); XDR (E, F). Figure 3 depicts the spectra obtained from locations denoted in Fig. 2 by arrows.

We would like to highlight two features of this applied approach, which make available the analysis of spectral features of interest: (i) the intensity of optical signal allows for catching the response not only from the cell wall but also from intracellular metabolites, *e.g.*, the nucleic acid content, and so on; (ii) collecting signals from closely spaced locations along the bacterial cell enables the identification of points where enhanced spectra are most clearly expressed.

For this purpose, the total range of wavenumbers 400–1,800 cm$^{-1}$ corresponding to the expected fingerprint region was subdivided for the subsequent analysis into parts within three spectral intervals: 400–1,000 cm$^{-1}$, 1,000–1,400 cm$^{-1}$, and 1,400–1,800 cm$^{-1}$. Figure 3 denotes this subdivision.

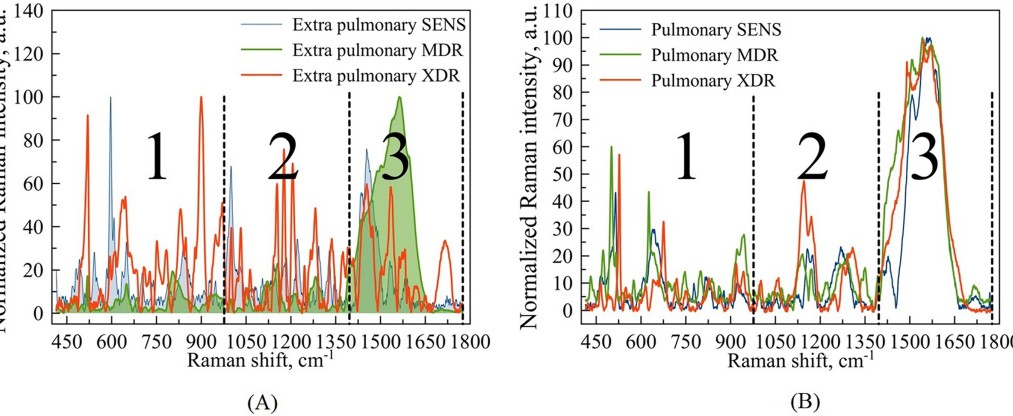

**Figure 3 Example of Raman spectra of extra pulmonary (A) and pulmonary (B) Mtb strains with different antibiotic resistance: SENS (blue line), MDR (green line), XDR (red line).** Wavenumber ranges are marked as large Arabic numbers. The numbers 1, 2, 3 correspond to the areas that were examined in more detail. The third area was not considered.

Based on our previous articles (*Zyubin et al., 2019*, *2021*), which operated with macroscopic, respectively to a cell's size and scale, we are focused on the regions denoted as 1 and 2 in Fig. 3 and do not consider the third region, since the latter contains mainly spectral bands associated with identifiers of proteins and amide groups, not evidently involved in the biochemistry of drug resistance.

Among the regions of interest, the most important is the spectral range $(720–800)$ cm$^{-1}$, which contains bands indicating DNA methylation signatures, as shown in Fig. 4A. It has been observed earlier, *e.g.*, see the arguments provided by *Papaleo et al. (2022)*, that DNA methylation plays a role in the formation of bacterial antibiotic resistance. Specifically, the intensity of the spectral bands at 747 and 775 cm$^{-1}$ can be corresponded to thymine base bands in probes of methylated DNA (*Kim et al., 2017*). In comparison with the results reported by *Kim et al. (2017)*, the second thymine peak (747 cm$^{-1}$) is shifted to 775 cm$^{-1}$ for XDR strains. Regarding the peaks associated with the methylation process, it is worth noting that a peak at 722 cm$^{-1}$ corresponds to the intermolecular vibrations of DNA/RNA, and its intensity is higher in XDR compared to MDR and SENS strains (Fig. 4A), which aligns with previous findings by *Kim et al. (2017)*. It is important to highlight that the peaks in the spectral range of $742–747$ cm$^{-1}$ observed in MDR and XDR strains may be attributed to thymine (*Kim et al., 2017*), but were not detected in SENS strains.

In the case of extrapulmonary samples (Fig. 5A), it is likely that a peak corresponding to adenine (*Stefancu et al., 2022*) in MDR and XDR strains can be identified at a frequency of 733 cm$^{-1}$, which slightly shifted relatively to the peak value (730 cm$^{-1}$) obtained in the work by *Stefancu et al. (2022)*. Concerned SENS strains it could be potentially shifted to 720 cm$^{-1}$ or closer to $742–740$ cm$^{-1}$ (see Fig. 5A).

The second narrow range $980–1,020$ cm$^{-1}$, presented in the Figs. 4B and 5B, is also of interest in terms of both DNA methylation markers (*Stefancu et al., 2022*) in resistance and phenylalanine, which is an amino acid that plays an important role in pathogen metabolism, see the work by *Griffin et al. (2012)*. Additionally, phenylalanine is also

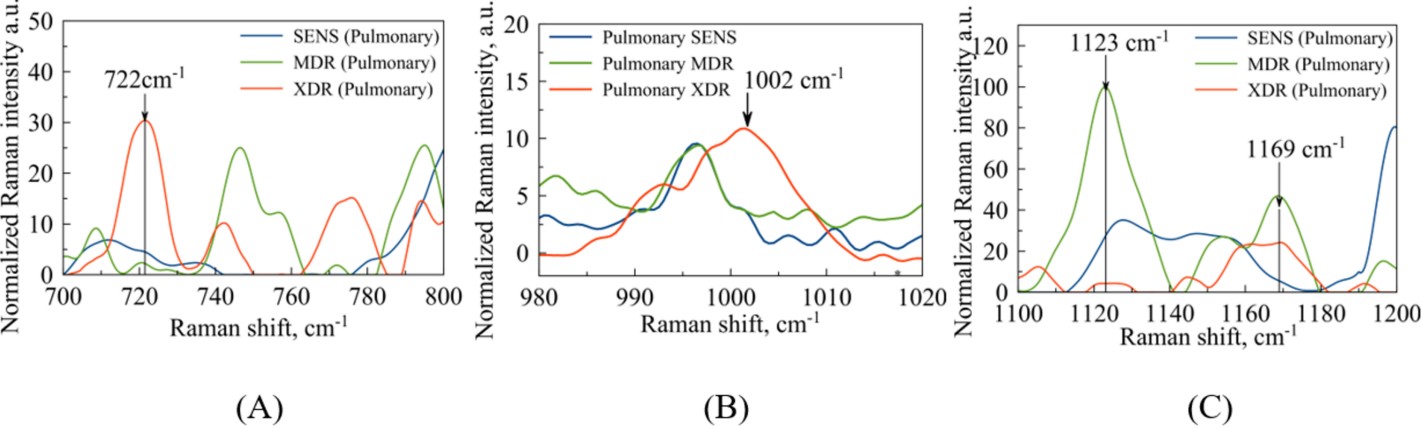

**Figure 4 Pulmonary Mtb strains with different antibiotic resistance: SENS (blue line), MDR (orange line), XDR (red line).** (A) Identified maxima in $700-800$ cm$^{-1}$ range: 722 cm$^{-1}$–purine/pyrimidine; 747 cm$^{-1}$–thymine; 772/776 cm$^{-1}$–cytosine/uracil; (B) identified maximum 1,002 cm$^{-1}$–phenylalanine (C) identified maxima in $1,100-1,200$ cm$^{-1}$ range: 1,123 cm$^{-1}$–DNA/RNA backbone; 1,169 cm$^{-1}$–glutathione.

involved in the regulation of gene expression and virulence factors in *M. tuberculosis* (*Gouzy, Poquet & Neyrolles, 2014*). Since the peaks of phenylalanine located at 1,002 cm$^{-1}$ (*Zyubin et al., 2019*) and 5-methylcysteine at 1,005 cm$^{-1}$ (*Stefancu et al., 2022*) are located quite close together, as seen in Fig. 4B, they can overlap for XDR strains. Interestingly, for strains with multiple drug resistance as well as for sensitive strains, these peaks are lower in intensity and significantly shifted to the left. As for extrapulmonary strains, as shown in Fig. 5B, the peak characteristics of phenylalanine are identical for all extrapulmonary strains.

Another demonstrable spectral feature is the Raman spectrum line situated around the frequency 1,170 cm$^{-1}$. It is clearly seen in Fig. 4C, the intensity of this band centred at 1,169 cm$^{-1}$ is close to zero for the sensitive strain, but noticeable for strains with antibiotic resistance.

When considering the SERS for whole cells (*Zyubin et al., 2021*), this line has been associated with glutathione (GSH) as corresponding to one of the principal indicator lines (*Qian & Krimm, 1994*) of this biomolecule involved in the host immune response to *M. tuberculosis* (*Allen et al., 2015*). Although mycobacteria do not synthesize glutathione themselves, its presence in clinical strains obtained from patients can be of exogenous nature since *M. tuberculosis* is, in principle, able to import GSH to its cytosol, see the respective discussion in the work by *Morris et al. (2013)*. In addition, it has been mentioned there that GSH is toxic for mycobacteria but one may expect that XDR strains can be safely at its higher concentration. The SERS spectral registration implemented in the present work is able to detect methylation of DNA, and this argues that other intracellular content can affect the Raman spectrum too.

It is also worth noting that *M. tuberculosis* can synthesize its own functional counterpart of GSH, mycothiol (MSH) protecting the bacterium from a stress action including that of a drug-induced one. Unfortunately, there are no available Raman spectra of this compound at the moment. But the SERS study by *Picot et al. (2019)* of N-acetyl-l-cysteine, which is

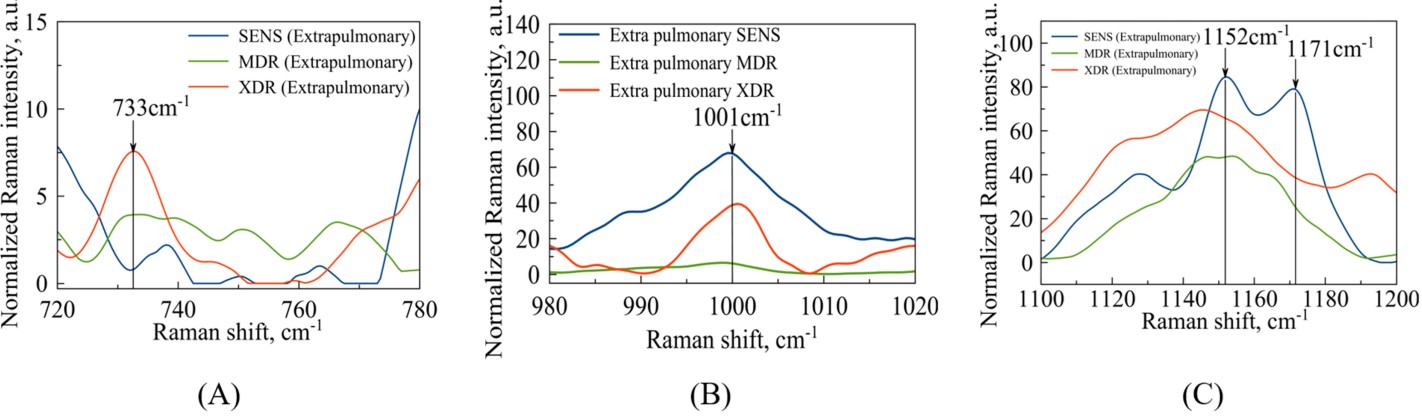

**Figure 5** **Extrapulmonary Mtb strains with different antibiotic resistance: SENS (blue line), MDR (orange line), XDR (red line).** (A) Identified maxima in $700-800$ cm$^{-1}$ range: 733 cm$^{-1}$–adenine; (B) identified maximum 1,001 cm$^{-1}$–phenylalanine (C) identified maxima in $1,100-1,200$ cm$^{-1}$ range: 1,152 cm$^{-1}$–N-H and C-H groups; 1,171 cm$^{-1}$–glutathione.

one of mycothiol's components, shows the existence of the band of $\delta C_\alpha H$ mode at the frequency 1,179 cm$^{-1}$ positioned not so far from the discussed one of GSH's. Thus, one should not discount the possibility of either alternative or combined origin of the drug resistance-indicating behaviour of the considered spectral band. However, any variant leads to the same visible manifestation demanded from the diagnostic point of view.

## Ensemble of individual cells

To explore whether the discussed features are general for spectra recorded from bacteria with different drug-resistance statuses, we carried out an additional analysis based on an ensemble of bacteria and TB types. This ensemble includes 77 spectra, among which 28 belong to the SENS class, 31 to the MDR class, and 23 to the XDR class; 36 and 41 spectra were taken from bacteria, which cause pulmonary and extrapulmonary TB types, respectively. For the full set of spectra and their properties, see the section 'Data Availability'.

As an additional test of the opportunity to use Raman spectra for solving the classification problem, here we use an alternative method of data processing. This approach relies on peak prominence, recently proposed as an effective method for handling spectra with complex backgrounds. It minimizes ambiguity that can arise from varying baseline removal techniques, see *e.g.*, the work by *Coca-Lopez (2024)*. Simultaneously, it is more reliable for searching positions of spectral peaks than their magnitude. Therefore, we applied this method implemented as a standard MATLAB function, determined a set of lines for each spectrum, and excluded as outliers the ones, which have prominence either less than half of the median prominence or a hundred times larger of it.

As the first issue to be investigated, we explored the possibility of distinguishing between different drug-resistance statuses using Raman spectral signatures. As such a signature, we used the number of peaks localized within 10 bins of 100 cm$^{-1}$ width centred from 450 to 1,450 cm$^{-1}$. The agglomerative hierarchical cluster tree analysis of these signatures was

carried out using the ward linkage method, *i.e.*, the minimum-variance algorithm operating with Euclidean distances, was applied. The resulting dendrogram giving the plot of the hierarchical binary cluster tree is shown in Fig. 6. One can see that indistinguishable clusters primarily comprise bacteria of a single type. Typically, the genomic variation between bacteria with different drug resistance status is smaller than the variation observed between distinct types of tuberculosis (TB). However, within a given TB type, significant differences may exist between clusters representing distinct drug-resistance statuses. Thus, we show that even this simplified line-counting approach could be used to solve the classification problem. This conclusion also confirms the idea of the significance of the Raman spectrum content preliminary discussed in the work by *Lavrova et al. (2023)* by the case study of a very limited number of single spectra.

As the second test, we explored the values of three particular Raman spectral lines discussed above. For this goal, we test whether there exists a detected maximum within the range of 10 $cm^{-1}$ centred in each of three frequencies considered as markers of drug resistivity. Their mean occurrences averaged over the ensemble are shown in Fig. 7. They indicate that there is a certain clearly expressed correlation between the detection of a maximum in all these ranges and the drug resistivity. Note that the applied method does not measure the peak's amplitude, *i.e.*, weakly expressed peaks make the same contribution in the resulting number of occurrences. This fact, along with potential shifts of adjacent peaks into the analyzed frequency range, may account for the presence of a maximum observed in approximately half of the drug-sensitive bacteria. However, growing drug resistance leads to more frequent detections. For the line 730 $cm^{-1}$, this dependence looks almost linear. The lines 747 $cm^{-1}$ and 1,170 $cm^{-1}$ behaves similarly and allow for identifying XDR cases. For the XDR case, the mean probabilities to detect three Raman lines are equal for all of them and higher than for alternatives. Thus, this ensemble-based picture supports the conclusion on their action as markers.

## Comparison of data for pathogen population and individual cells

In bacterial populations or conglomerates, characteristic peaks have already been found, such vibrational modes related to the DNA and RNA backbone at 1,123 $cm^{-1}$ were also identified (*Dhankhar, 2018*) for MDR pulmonary strains at Fig. 4C. The Raman peaks located at shift position 1,152 $cm^{-1}$ represented a cell wall's N-H, C-H in-plane bendings were also identified (*Ullah et al., 2022*) for sensitive extrapulmonary strains at Fig. 5C. Separately, it is worth noting that the vibrations corresponding to purines and pyrimidines of DNA at 722 $cm^{-1}$ were successfully identified (*Zyubin et al., 2019*) for XDR pulmonary strains as in Fig. 4A. The cytosine and uracil band modes at 772/776 $cm^{-1}$ were discovered in the article (*Zyubin et al., 2019*), for XDR Pulmonary strains at Fig. 4A.

It should be noted that the peaks of pyrimidine and purine bases, as well as individual bases related to DNA, are manifested mainly in MDR and XDR strains and are completely absent in sensitive strains. This fact requires further and more detailed consideration, such as the identification of metabolic features of such mycobacteria.

However, another marker thymine in DNA was found for individual bacteria at the wavelength 747 $cm^{-1}$, which is characteristic for MDR pulmonary strains. Nevertheless,

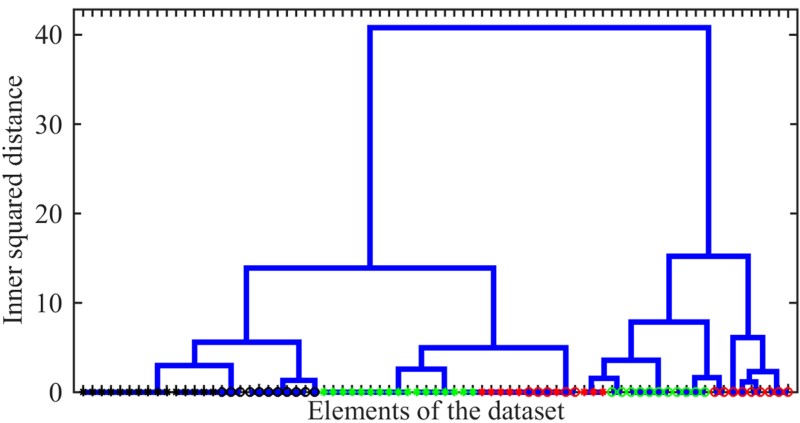

**Figure 6 The hierarchical binary cluster tree for the investigated ensemble of 77 Raman spectra.** Green, black, and red colours indicates SERS, MDR and XDR bacteria, respectively. Circles and asterisks denote pulmonary and extra pulmonary TB types.

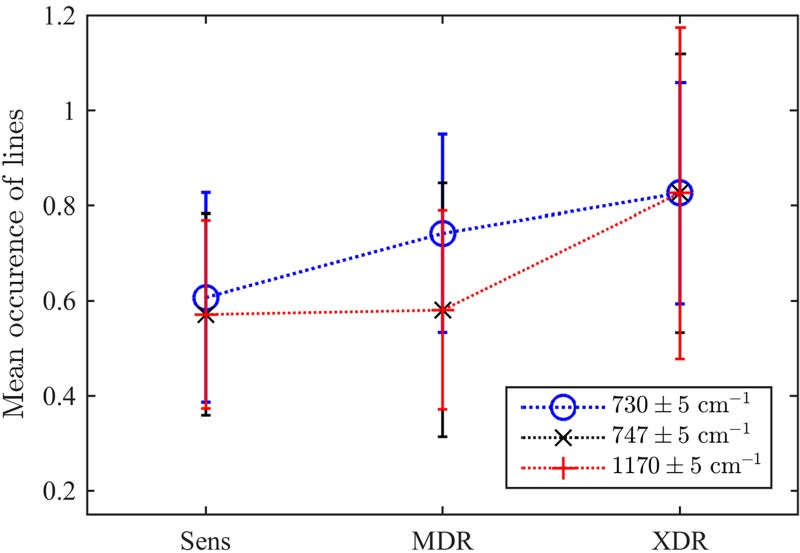

**Figure 7 The ensemble-averaged probability to detect a peak within the range designated in the legend for bacteria with different drug resistance statuses; error bars denote the 95% confidence intervals; dotted lines connect markers for visual guidance.**

overall, although more detailed information about specific marker peaks can be revealed on individual bacteria, the main information about differences between strains can also be obtained on conglomerates.

In principle, the differences between conglomerates and individual bacteria are based on the heterogeneity of the culture being studied. When studying conglomerates, average values of molecular structure peaks are identified, whereas, in individual bacteria, individual peaks may not be detected or may be greatly shifted depending on the state of the cell itself, in particular on the "age" of the cell. Nevertheless, the overlapping peaks of phenylalanine and 5-methylcytosine found in XDR lung strains that coincide with

conglomerates (*Zyubin et al., 2019*) may serve as markers for strains with extra drug resistance. It should be pointed out that attention to a possible interplay between 5-methylcytosine and drug resistance as involved in the process of methylation of certain cytosine residues, has been drawn a long time ago (see *Hattman, Gold & Plotnik, 1972*) and considered now as one of the promising methylation markers, which can be detected by Raman spectroscopy (*Li et al., 2018*; *Zhang et al., 2023*). Concerning the case of *M. tuberculosis*, an epigenetic mechanism, which plays a significant role in DNA methylation mediated by 5-methylcytosine has been analysed recently using the methylation motif analysis by *Phelan et al. (2018)* and with metabolomic tracing by *Gong et al. (2021)*. Thus, our results support this interpretation with results that can be achieved by more simple instrumentation.

## CONCLUSION AND OUTLOOKS

The principal results of this work consist of a demonstration of the possibility of accessing molecular features, which distinguish between drug-sensitive and drug-resistant *Mycobacterium tuberculosis* by the adjustment of spatial registration of the surface-enhanced Raman signal respectively to a bacterium's body. The number of trial locations, as shown using a 785 nm diode laser and deactivated *M. tuberculosis* strains belonging to the Beijing family, can reach up to six distinct points.

Exploring the "fingerprint" region of $400-1,800$ cm$^{-1}$, we identified spectral bands of 730 cm$^{-1}$, 747 cm$^{-1}$, and $1,170 \pm 2$ cm$^{-1}$ as the most promising to be biomarkers of drug resistance. The first two bands are associated with DNA methylation patterns involving 5-methylcytosine and the methylated thymine group. The last band is considered to correspond to the typical Raman band known for glutathione, which is involved in the immune response to mycobacteria and is known as an attractive target for the SERS-based analysis, as shown in *Ma & Huang (2015)*. The respective spectral maxima either at $1,169 \pm 2$ cm$^{-1}$ or at $1,171 \pm 2$ cm$^{-1}$ are extremely well expressed in the cases of pulmonary and extrapulmonary XDR strains. At the same time, this demonstrable biomarker may be also related to the mycobacterial counterpart of glutathione, mycothiol, and this fact provides an outlook for future SERS-based studies of this mycobacterium-specific compound.

Thus, we would like to highlight that the proposed method looks as having certain advantages over conventional approaches used to establish a correspondence between drug resistance status and phenotype of *M. tuberculosis*. In particular, it opens some perspectives for exploring the fundamental question of which features of a cell's molecular architecture may be chosen as potential targets for developing new drugs aimed at fighting the emergent drug resistance.

Summarising, we can conclude that SERS measurements, which are characterised by a high speed of operation and accuracy, can be a very efficient tool for fundamental biophysical research of molecular biophysics of mycobacterial cells. In addition, since the respective experimental procedure takes less than 1 h, it also may find clinical applications when one needs a fast characterization of the sample since the conventional procedures, as a rule, require several days. Such molecular methods as various types of the quantitative

real-time polymerase chain reaction (PCR), see, *e.g.*, the reviews by *Palomino (2009)*, *Schön et al. (2017)*, *Mugenyi et al. (2024)*, address, first of all, the genetic origin of resistance to specific drugs with the known mechanism of action. Simultaneously, evidence is emerging that one of the critical factors impeding drug efficacy is the thickening of the bacterial cell wall, accompanied by alterations in its molecular composition. These changes can arise in response to environmental interactions (for a review of recent findings, see recent state-of-the-art studies by *Schami et al., 2023*; *Shukla, Bhardwaj & Singh, 2024*). In this case, one needs methods of physical chemistry, which directly operate with features of molecular rheology (*Postnikov et al., 2023*). The single-cell-based SERS is the natural choice for such a goal since it originally operates with specific molecular bond oscillations and allows the characterization of effects of molecular packing and spatial localization of specific markers. Apart of the fundamental significance, this approach will allow such a characterization of particular bacteria extracted from patient samples in clinical conditions.

At the same time, the reported results, which represent a pilot study, have some limitations. Although we achieved data sampling from different locations on the single cell's surface, the comparative analysis of Raman spectral features respective to each particular position remains the task for future works following, *e.g.*, the strategy considered for other types of probes by *Ilchenko et al. (2024)*. Another perspective future problem is an analysis of the correspondence between the genotypic and phenotypic variations in drug-resistant mycobacteria as has been studied, for example, for *H. pylori* by *Liu et al. (2022)*.

In addition, one has to note that one molecule exhibits multiple Raman spectral lines at different frequencies. At the moment, we operated mainly with the most expressed lines for each marker compounds. Therefore, there is interest in future studies to improve the analysis operating with sets of a compound's characteristic lines simultaneously. Finally, such identifications may be further compared with the results of molecular simulations.

### Funding
This research was supported by the Ministry of Science and Higher Education of the Russian Federation, project 075-02-2024-1430. The funders had no role in study design, data collection and analysis, decision to publish, or preparation of the manuscript.

### Grant Disclosures
The following grant information was disclosed by the authors:
Ministry of Science and Higher Education of the Russian Federation: 075-02-2024-1430.

### Competing Interests
The authors declare that they have no competing interests.

## Author Contributions

- Andrey Zyubin conceived and designed the experiments, analyzed the data, prepared figures and/or tables, authored or reviewed drafts of the article, and approved the final draft.
- Anastasia Lavrova conceived and designed the experiments, analyzed the data, prepared figures and/or tables, authored or reviewed drafts of the article, and approved the final draft.
- Marine Dogonadze performed the experiments, analyzed the data, prepared figures and/or tables, and approved the final draft.
- Evgenii Borisov performed the experiments, prepared figures and/or tables, and approved the final draft.
- Eugene B. Postnikov conceived and designed the experiments, analyzed the data, prepared figures and/or tables, authored or reviewed drafts of the article, and approved the final draft.

## Human Ethics

The following information was supplied relating to ethical approvals (*i.e.*, approving body and any reference numbers):

Federal State Budgetary Institution "St. Petersburg Research Institute of Phtisiopulmonology" of the Ministry of Health of The Russan Federation.

## Data Availability

The raw spectral data and the code used for the data processing and examples of the raw data are available at GitHub and Zenodo:

- https://github.com/postnicov/RamanMTBbaseline.
- Zyubin, A. (2025). The MATLAB code used for processing of SERS spectra of individual micobacteria. Zenodo. https://doi.org/10.5281/zenodo.14608082.

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
