# Peer review of "Single-cell analysis of Mycobacterium tuberculosis with diverse drug resistance using surface-enhanced Raman spectroscopy (SERS)"

_PeerJ, doi:10.7717/peerj.18830_

## Round 0.1 · original submission · Major Revisions

Your work has now been assessed by 3 independent reviewers and all of them agree to the recommendation of including additional controls, improving the quality of figures and clarity in some paragraphs.Therefore I would like to encourage you to revise your work after these recommendations and respond to all criticisms.

**Language Note:** The review process has identified that the English language must be improved. PeerJ can provide language editing services - please contact us at [email protected] for pricing (be sure to provide your manuscript number and title). Alternatively, you should make your own arrangements to improve the language quality and provide details in your response letter. – PeerJ Staff

Reviewer 1 ·

Basic reporting

The authors conducted a label-free (gold nanoparticles) SERS mapping analysis at the single-cell level for Mtb strains with three groups of antibiotic resistance profiles: SENS, MDR, and XDR. In addition, the authors also investigated the differences between extra-pulmonary and pulmonary Mtb strains. According to the study, the authors identified unique features in glutathione bands and DNA methylation patterns that may be linked with drug sensitivity for Mycobacterium tuberculosis strains. Therefore, the authors concluded that untargeted single-cell SERS analysis could be a promising tool for antibiotic resistance identification, which may contribute to adequate therapy of Mtb infections (precision medicine). In general, the study is very interesting since it uses single-cell SERS to identify potential components or modifications in the cell wall structure to explain the variances for Mtb drug sensitivity and resistance. In addition, the pilot study also developed a potential method for distinguishing Mtb with different antibiotic resistance profiles, facilitating the application of the method in clinical settings. The reviewer sees novelty and significance in the study. However, to qualify for further consideration, some major revisions still need to be made. For details, please refer to the comments below.
1. English must be improved by a native speaker, a professional language service, or at least language-checking software such as Grammarly to check all typos and grammar mistakes.
2. There are still some important literature references that need to be cited to make the background/context provide sufficiency, e.g., 10.1038/s41467-019-12898-9
3. Article structure is okay. However, tables and figures need to be improved for better demonstration. I'll give you the details below.
4. Results are okay to support the hypothesis, that is, the application of SERS to accurately discriminate Mtb with different antibiotic resistance profiles. In addition, a potential molecular mechanism (component or modification) is given to explain the reason.

Experimental design

There are some questions about the experimental design. I would like to see how the authors conduct these experiments in detail.
1. The sample size for each category is small, as shown in Table 1 (SERS, 2; MDR 3; XDR 1), which makes me worry about the reliability of the results and conclusion.
2. Although single-cell SERS will increase the study's accuracy, reproducibility is still required for each sample. That is, how many SERS spectra were acquired for each strain? Is there a standard error for all the SERS spectra of each strain so consistency can be analyzed for the experiment?
3. Why did the authors choose gold nanoparticles over copper or silver nanoparticles? Has any experiment been conducted to compare the performance of silver vs. gold nanoparticles?
4. Why did the authors want to compare the extra-pulmonary and pulmonary Mtb strains for their antibiotic resistance differences in SERS analysis? Please cite previous studies to validate the experiment design. Is there any reference to mention that Mtb strains from the two locations have different antibiotic resistance profiles or different resistance mechanisms?

Validity of the findings

I have the following concerns in terms of the validity of the findings. The authors need to respond to these concerns with adequate evidence.
1. Can the authors provide an in-depth analysis of their findings with literature, such as why 5-methylcytosine is related to antibiotic sensitivity and resistance in Mtb. Is there any other evidence to support the link?
2. I would suggest that for each Mtb type, at least 3 strains must be tested for label-free single-cell SERS analysis to make the results statistically meaningful.
3. Clustering analysis is strongly recommended for the authors to see the intrinsic differences among the Mtb strains because SERS spectra are too complex to be analyzed via the naked eye. Multiple quantitative methods must be conducted to validate the similarities and differences among these spectra.

Additional comments

The authors should also consider the following suggestions to improve the manuscript.
1. Figures 1 and 2, except for using colors to designate the strains, please also add text in the figure to show the names of the strains (SENS, MDR, XDR). Figure 1A's figure legend is too vague. What does the color mean for the four lines?
2. Figure 3. A and B should be extra-pulmonary and pulmonary. The figure legend is not correct. Please revise it.
3. Figure 4 and 5. Blank space needs to be used consistently. A. 722cm-1. B. 1002 cm-1. Please revise it.
4. Although it is okay to mix results with discussion, the discussion part needs more insights. Please compare the method, study design, and results with other published studies to justify the novelty and significance of your study, especially in clinical settings. For example, when compared with mass spectrometry or qPCR, why does label-free single-cell SERS have advantages?
5. The limitation of the study needs to be specified.

·

Basic reporting

The writing is clear.
I recommend some references below, just as a reference for data interpretation.
An improvement in your micrographs is recommended.

Experimental design

The experimental design is adequate, it is only suggested that the appropriate controls be shown, to have a greater strength of the discussion.

A description of a methodology is suggested.

Validity of the findings

The article reports the analysis of SERS/RAMAN signals of Mycobacterium tuberculosis strains of the Beijing family of TB with different drug sensitivities, as a possible diagnostic strategy. The study is based on the SERS methodology previously published by this group.

This is an interesting article that may be useful for the design of SERS-type biosensors aimed at diagnosing an effective treatment.

However, it is recommended to follow the following suggestions.

1. Table 1 lists the clinical strains that were used indicating three strains for MDR and one for XDR, however, in Fig 2, two strains for MDR and two for XDR are analyzed. Could the variation be due to a typographical error in the table?
2. It would be advisable to describe the SERS preparation of the samples in the methods section.
3. It would be recommended to modify the microscopic images in order to have a better visibility of the bacteria, especially in the image in Fig 1.
4. In the spectra of Fig. 1, it is observed that the peaks vary between each reading zone, so I suggest first eliminating the reading slope in order to better appreciate the peaks generated by the compounds and then discuss the variations.
5. Likewise, it is recommended to perform the optimization of the total spectra of the bacteria, because the RAMAN analysis is not always reported on the same area in each type of bacteria (SENS, MDR and XDR), which means that the variations may not be attributed only to the type of pharmacological sensitivity. It is recommended, if possible, to manage a SERS mapping throughout the bacteria (https://doi.org/10.1038/s41467-024-47044-7 )
6. It would be desirable to show the controls used to corroborate the observations of compounds not belonging to the cell wall, such as internal metabolites and especially methylations at the genetic level as part of the fingerprint. This is because it is observed that there is a variability of signal throughout the same bacteria. If you like, as supplementary material.
7. Finally, I suggest that the following references be taken into account.
Biophysical Journal 114, 2498–2506, June 5, 2018 , Talanta 253 (2023) 123941, Results in Physics 44 (2023) 106106, RSC Adv., 2015,5, 57847-57852, Analyst, 2022, 147, 4674

Reviewer 3 ·

Basic reporting

The authors have written a clear and concise description of the presented work. However, the citations are not properly integrated, causing the sentence structure to be a bit choppy at times. For example, " the authors of the work Wang et al. (2022) among other results of the classification problem, proposed a set of bandwidth.." is an incomplete phrasing.

Other major concerns include:

1. Figure 1 A and B are low-resolution images that are a bit blurry, with certain parts of the graph being cut off or missing. For example, the cyan-labeled spectra on the graph are incomplete. Also, the SERS spectra displayed in Figure 1A showcase more background auto-fluorescence than a true SERS signal.

2. Figure 2 is also blurry, and it is difficult to see the bacteria structure being mapped.

Experimental design

Although the experimental design is well-defined, mapping the entire cell from cell wall to cell wall would be ideal for assessing the biomolecular differences across the entire bacteria for each drug-resistant strain. The multiple-point measurement does not depict a true map of the specimen. Or more coverage for the point measurements are needed.

Validity of the findings

The SERS spectra displayed in Figure 1A showcase more background auto-fluorescence than a true SERS signal. The fluorescence-subtracted spectra in Figure 3, unfortunately, have a very low signal, which also negates it being a surface enhancement of the Raman signal, highlighting the limitations of this research.

Also, information about the number of bacterial cells taken per TB type and per drug resistance is missing to validate the SERS signal obtained further.

The authors used well-known literature to identify well-characterized and acceptable peaks. However, since the spectra are within the signal's noise bed, it is challenging to determine the validity of the observed peaks.

---

## Round 0.2 · Minor Revisions

There are some comments from reviewer 1 that would surely improve your work, therefore I encourage you to revise accordingly.

Reviewer 1 ·

Basic reporting

In general, the manuscript reported an interesting and advanced study using Raman spectroscopy to detect Mycobacterium tuberculosis drug resistance at a single-cell level. Although the study itself is interesting and holds novelty in the field (there are also other similar studies, though, in different bacterial pathogens, e.g., Helicobacter pylori. Clinical Chemistry, 2023), the manuscript still requires major revisions before it can be considered for further review. Most importantly, the language of the manuscript needs a major revision. There are many typos, errors, and language misuse. For example, Mycobacterium tuberculosis is abbreviated as Mtb, not Mbt. By the way, when using Latin names of the bacterium, please italicize it. Thank you.

Experimental design

The experimental design is qualified in general. However, it is not good when focusing on details of the experimental procedures, especially when talking about Raman spectral analysis such as how the characteristic peaks were identified and why they were related to particular peaks, etc. In addition, the single-cell procedure was also vaguely described in the study, which needs to be provided in detail, especially about how to use the Raman spectrometer for generating single-cell spectra.

Validity of the findings

The findings of the study is generally meaningful. However, through single-cell analysis, the authors claim that they can identify antibiotic resistance and sensitivity. Considering the large amount of cells in the samples, how the authors can make sure that their results are representative? What is the throughput of the analysis? Can it be high-throughput so that as many Mtb cells can be analyzed within a limited time as possible so that the results can contribute to the clinical interpretation?

By the way, considering the high inconsistency between genotype and phenotype, can the authors use the SERS technique to address this issue?

I am also interested to ask the authors, as for the pulmonary and extrapulmonary Mtb infections, is there any treatment differences in terms of antibiotic dosage?

Considering the MIC of Mtb eradication, can the SERS technique be used to correlate with MICs of Mtb Strains? Please comment. The answer to this question is related to the clinical application of the SERS technique.

Additional comments

Please revise your manuscript thoroughly and provide point-by-point responses to the reviewer's comments with clarity.

·

Basic reporting

The basic reporting are adequate and the observations were addressed.

Experimental design

The experimantal design are adequate and the observations were addressed.

Validity of the findings

The validity of the findings are adequate and the observations were addressed.
The document is detailed and supplemented with the changes made.

---

## Round 0.3 · accepted · Accept

I am thankful for your having taken into account the recommendations raised by the reviewers.

Congratulations on the acceptance of your work for publication in PeerJ!

Reviewer 1 ·

Basic reporting

The manuscript is well-written in clear and professional English. The structure of the study is rational and the figures and tables are well presented. Raw data and codes are available via GitHub at https://github.com/postnicov/RamanMtbbaseline. The literature is sufficient and up-to-date.

Experimental design

The research falls into the aims and scopes of the journal PeerJ. The research question is well-defined and is meaningful with potential real-world applications, especially when considering that Mtb is still pandemic in the world and should be diagnosed with simplicity, rapidity, and accuracy. Methods described in the study are sufficient.

Validity of the findings

Conclusions are well-stated and linked to the original research question. Limitations of the study were also emphasized, which requires future experimental and computational studies in the field.

Additional comments

No comments